# Relationship between high shear stress and OCT-verified thin-cap fibroatheroma in patients with coronary artery disease

**Naotaka Okamoto[1], Yuliya Vengrenyuk[1], Valentin Fuster[1], Habib Samady[2], Keisuke Yasumura[1], Usman Baber[1], Nitin Barman[1], Javed Suleman[1], Joseph Sweeny[1], Prakash Krishnan[1], Roxana Mehran[1], Samin K. Sharma[1], Jagat Narula[1], Annapoorna S. Kini[1]***

**1** Division of Cardiology, Mount Sinai Hospital and Icahn School of Medicine at Mount Sinai, New York, NY, United States of America, **2** Division of Cardiology, Department of Medicine, Emory University School of Medicine, Atlanta, GA, United States of America

* annapoorna.kini@mountsinai.org

**Data Availability Statement:** All relevant data are within the manuscript and its Supporting Information files.

## Abstract

High-risk coronary plaques have been considered predictive of adverse cardiac events. Both wall shear stress (WSS) in patients with hemodynamically significant lesions and optical coherence tomography (OCT) -verified thin-cap fibroatheroma (TCFA) are associated with plaque rupture, the most common underlying mechanism of acute coronary syndrome. The aim of the study was to test the hypothesis that invasive coronary angiography-based high WSS is associated with the presence of TCFA detected by OCT in obstructive lesions. From a prospective study of patients who underwent OCT examination for angiographically obstructive lesions (Yellow II), we selected patients who had two angiographic projections to create a 3-dimensional reconstruction model to allow assessment of WSS. The patients were divided into 2 groups according to the presence and absence of TCFA. Mean WSS was assessed in the whole lesion and in the proximal, middle and distal segments. Of 70 patients, TCFA was observed in 13 (19%) patients. WSS in the proximal segment ($WSS_{proximal}$) (10.20 [5.01, 16.93Pa]) and the whole lesion ($WSS_{lesion}$) (12.37 [6.36, 14.55Pa]) were significantly higher in lesions with TCFA compared to $WSS_{proximal}$ (5.84 [3.74, 8.29Pa], p = 0.02) and $WSS_{lesion}$ (6.95 [4.41, 11.60], p = 0.04) in lesions without TCFA. After multivariate analysis, $WSS_{proximal}$ was independently associated with the presence of TCFA (Odds ratio 1.105; 95%CI 1.007–1.213, p = 0.04). The optimal cutoff value of $WSS_{proximal}$ to predict TCFA was 6.79 Pa (AUC: 0.71; sensitivity: 0.77; specificity: 0.63 p = 0.02). Our results demonstrate that high WSS in the proximal segments of obstructive lesions is an independent predictor of OCT-verified TCFA.

## Introduction

Wall shear stress (WSS) is the tangential force produced by the luminal blood flow on the vascular intima. Whereas physiological WSS facilitates atheroprotective signals, low WSS contributes to endothelial inflammation and proatherogenic milieu, and high WSS leads to matrix

**Funding:** The author(s) received no specific funding for this work.

**Competing interests:** I have read the journal's policy and the authors of this manuscript have the following competing interests: Dr. Samady has received research grants from Abbott Vascular, Medtronic, National Institutes of Health, St. Jude Medical, and Gilead. Dr. Baber: speaker honoraria from Boston Scientific and Amgen, speaker honoraria and grants from AstraZeneca; Dr. Mehran: Abbott Vascular consultant and research grant, Boston Scientific consultant. Dr. Sharma: speaker honoraria from Abbott, Boston Scientific, Cardiovascular Systems, Inc. All other authors have no relationships relevant to the contents of this paper. This does not alter our adherence to PLOS ONE policies on sharing data and materials.

metalloproteinase activation with attenuation of fibrous cap and enlargement of necrotic core with expansive remodeling [1–5]. What is not very clear is whether high WSS induces high-risk plaques feature or vice versa. Nonetheless, high WSS may underlie plaque rupture or erosion and result in an acute coronary event [6–8]. In patients with hemodynamically significant lesions from FAME II trial, angiography-based higher WSS in the proximal segments of the lesions was a predictor of subsequent myocardial infarction (MI) within 3 years [9]. Correlation between angiography-based WSS and fibrous cap thickness in obstructive lesions has not been previously described. In the present study, we identified angiography-based WSS using computational fluid dynamics (CFD) and correlated it to OCT-verified thin-cap fibroatheroma (TCFA) in obstructive lesions from patients with coronary artery disease (CAD). We hypothesized that the angiography-based high WSS will be associated with the presence of high-risk plaques detected by OCT.

## Materials and methods

### Study population and design

The protocol of YELLOW II study was approved by the Institutional Review Board of the Mount Sinai School of Medicine. All patients provided written informed consent. YELLOW II study design and results have been previously described [10]. Briefly, in the prospective study, 85 patients with multivessel stable CAD requiring staged percutaneous coronary intervention (culprit lesion initially, obstructive non-culprit lesion later) underwent multimodality intravascular imaging with OCT, intravascular ultrasound (IVUS) and near-infrared spectroscopy (NIRS) of the obstructive non-culprit lesion before and after 8–12 week intensive statin therapy. There were 85 lesions in 85 patients; all lesions were stented during follow-up procedure. 85 patients, 70 patients were selected who had 2 angiographic projections at least 25 degree apart at baseline to allow accurate coronary geometry reconstruction and WSS assessment. The patients were divided into 2 groups according to the presence and absence of TCFA.

### OCT and NIRS/IVUS image acquisition and analysis

OCT examination was performed using C7-XR$^{TM}$ OCT Intravascular Imaging System and Dragonfly$^{TM}$ imaging catheter (Abbott Vascular, Santa Clare, CA). The OCT catheter was inserted at least 10 mm distally to the study target lesion. Commercially available TVC (True Vessel Characterization) imaging system with the TVC insight catheter (Infraredx, Burlington, Massachusetts) was used to acquire combined NIRS and IVUS image for the same lesion.

OCT images were analyzed according to the current consensus standards [11] as previously described in YELLOW II study [10]. Lipid core was identified as a signal-poor region with poorly delineated borders, little or no signal backscattering, and an overlying signal-rich layer, the fibrous cap (S1 Fig). Lipid arc was measured at 1-mm interval to obtain the maximum and average values and calculate lipid volume index as a product of the average lipid arc and lipid length. Fibrous cap thickness (FCT) was measured 3 times at its thinnest part and the average value was calculated. OCT defined lipid rich plaque (LRP) was defined as lipid plaque with maximal lipid arc more than 90˚. TCFA was defined as a LRP with FCT less than 65 μm.

Raw spectroscopic information from NIRS chemograms was transformed into a probability of lipid that was mapped to a red-to-yellow colour scale, with a low probability of lipid shown as red and a high probability of lipid shown as yellow (S1 Fig). Lipid core burden index (LCBI) was calculated by dividing the number of yellow pixels (probability $\geq 0.6$) by the total number of viable pixels within the plaque multiplied by 1000. Maximal LCBI for the whole lesion (maxLCBI$_{lesion}$), within any 10-mm (maxLCBI$_{10mm}$) or 4-mm segment (maxLCBI$_{4mm}$) were calculated for each lesion [12]. IVUS images were analyzed off-line using computerized

planimetry software (echoPlaque 4.0, INDEC Medical Systems, Inc, Santa Clara, CA) according to the current guidelines as previously described [12, 13]. Quantitative analysis included measurements of the external elastic membrane (EEM) and lumen cross-sectional areas (CSA) at 1-mm interval. Plaque+media CSA was calculated as EEM minus lumen CSA, and plaque burden was calculated as [plaque+media/EEM CSA*100]. Simpson's rule was used to estimate EEM volume and total atheroma volume (TAV); percent atheroma volume (PAV) was calculated as TAV divided by EEM volume. OCT and NIRS/IVUS image analysis was performed at the Cardiovascular Research Foundation (New York, NY), which had no knowledge of patients clinical presentation.

## Angiographic reconstruction and computational fluid dynamics

Three-dimensional quantitative coronary angiography (3D-QCA) and vessel reconstructions were performed using two end-diastolic projections at least 25˚ apart with commercially available software QAngio XA 3D RE (Medis, Leiden, The Netherlands) followed by automatic quantification of 3D lesion length, minimal and reference diameters, and percent diameter stenosis (DS) (Fig 1A) [14]. Patient-specific vessel geometries were exported to ANSYS Fluent 19.0 (ANSYS Inc. Pennsylvania, USA) for CFD simulations (Fig 1B). Patient-specific inlet velocities were calculated using frame counts ([segment length assessed by angiography]* [acquisition speed]/[last frame number–first frame number]) and applied as inlet boundary conditions with a flat profile as previously described [9]. For bifurcation lesions, outlet boundary conditions were computed according to Murray's law [15]. The density and viscosity of blood were set to 1,050 kg/m$^3$ and 0.0035kg/m·sec respectively. No-slip boundary condition was applied at the vessel wall and blood flow was assumed to be steady. Each angiography-defined lesion was divided into 3 equal parts: proximal, middle and distal following the segment definitions introduced in the WSS FAME II sub-study [9]. Upstream and downstream segments were defined as segments proximal or distal to the lesion. Mean and maximal WSS were calculated for each segment and the whole lesion and compared between TCFA and non-TCFA lesions detected by OCT (Fig 1C). The optimal mesh size was determined based on a mesh refinement study to ensure that the results are not affected by the element size. The average number of finite elements in the study was 287,265, and the mean surface area was 0.021 mm$^2$. Three-dimensional geometry reconstruction and CFD calculations were done at the Mount Sinai Intravascular Imaging Laboratory (New York, NY).

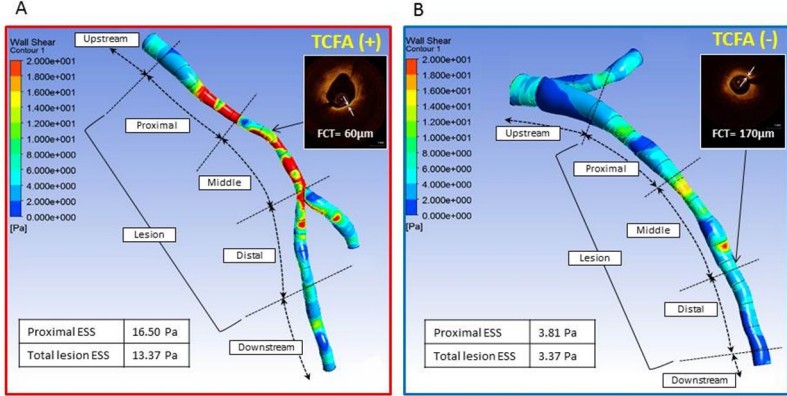

**Fig 1. Angiography-based assessment of wall shear in obstructive coronary lesions.** CFD analysis was performed in models reconstructed from two angiographic projections (A); Mean WSS was calculated in the whole lesion and three lesion sub-segments: proximal, middle and distal. OCT imaging was performed in the same lesion to identify TCFA lesions.

## Statistical analysis

Continuous variables are presented as a mean ± standard deviation or median (interquartile range) depending on data distribution. The data was compared with t-test or Mann-Whitney U test. Categorical variables are presented as frequency counts and percentages and were compared with chi-square test or Fisher exact test as appropriate. Logistic regression analysis was performed to detect the predictors of the presence of TCFA. A receiver operating characteristics curve was used to detect the optimal cutoff value for detecting TCFA. P value less than 0.05 was considered to be statistically significant and all statistical analyses were performed using SPSS 24.0 (IBM Corp. Armonk, NY).

## Results

### Patient characteristics

OCT-verified TCFA was observed in 13 (19%) of the patients. Table 1 summarizes patient baseline characteristics according to the presence or absence of TCFA lesion. Age, gender, the prevalence of hypertension, hypercholesterolemia and diabetes mellitus were comparable for the 2 groups. Statin was similarly used and laboratory data including LDL-cholesterol were not different between the groups.

### OCT and NIRS/IVUS findings

OCT-verified minimum fibrous cap thickness was 60 μm (interquartile range [IQR]: 50 to 60μm) in TCFA lesions and 100 μm (IQR: 90 to 130μm, p<0.01) in lesions without TCFA (Table 2). TCFA lesions had a higher prevalence of LRP (100%) compared to no TCFA group (73.7%, p = 0.03). Lipid content including lipid arc maximum, lipid length and lipid volume index were higher in TCFA lesions compared lesions without TCFA. The prevalence of

**Table 1. Patient baseline characteristics.**

|  | TCFA (n = 13) | No TCFA (n = 57) | P-value |
|---|---|---|---|
| Age | 63.69 ± 11.45 | 63.12 ± 10.34 | 0.86 |
| Male | 7 (53.8) | 39 (68.4) | 0.25 |
| Hypertension | 11 (84.6) | 51 (89.5) | 0.32 |
| Hypercholesterolemia | 10 (76.9) | 50 (87.7) | 0.17 |
| Diabetes mellitus | 6 (46.2) | 24 (42.1) | 0.87 |
| Current smoking | 1 (7.7) | 11 (19.3) | 0.25 |
| Prior MI | 1 (7.7) | 8 (14.0) | 0.45 |
| Prior PCI | 0 (0) | 19 (33.3) | 0.01 |
| Statin use | 10 (76.9) | 47 (82.5) | 0.45 |
| Coronary vessel |  |  | 0.29 |
| LAD | 5 (38.5) | 25 (43.9) |  |
| LCX | 2 (15.4) | 18 (31.6) |  |
| RCA | 6 (46.2) | 14 (24.6) |  |
| Total cholesterol, mg/dl | 151.0 (127.0, 188.5) | 137.5 (144.8, 165.5) | 0.22 |
| LDL cholesterol, mg/dl | 84.8 (54.1, 116.6) | 74.8 (55.9, 91.3) | 0.39 |
| HDL cholesterol, md/dl | 41.0 (32.5, 54.0) | 39.5 (33.0, 47.5) | 0.82 |
| Triglyceride, mg/dl | 122.0 (77.5, 156.5) | 99.0 (63.0, 144.3) | 0.29 |

Values are mean ± SD, median (interquartile range) or n (%); TCFA = thin-cap fibroatheroma (cap ≤65μm); LAD = left anterior descending artery; LCX = left circumflex coronary artery; RCA = right coronary artery; LDL = low-density lipoproteins; HDL = high-density lipoproteins.

**Table 2. Intravascular imaging data.**

| | TCFA (n = 13) | No TCFA (n = 57) | P-value |
|---|---|---|---|
| **OCT** | | | |
| Minimum lumen CSA, mm$^2$ | 1.64 ± 0.69 | 1.92 ± 0.69 | 0.19 |
| Lipid rich plaque | 13 (100) | 42 (73.7) | 0.03 |
| Lipid arc maximum, ˚ | 189.2 ± 66.7 | 134.3 ± 74.9 | 0.02 |
| Lipid arc average, ˚ | 123.3 ± 37.3 | 95.8 ± 54.2 | 0.04 |
| Lipid length, mm | 8.4 (7.3, 10.4) | 4.6 (2.3, 7.1) | < 0.01 |
| Lipid volume index, ˚ x mm | 1009.6 (663.5, 1296.0) | 432.7 (215.9, 774.2) | < 0.01 |
| Minimum fibrous cap thickness, μm | 60 (50, 60) | 100 (90, 130) | < 0.01 |
| Macrophage | 13 (100) | 57 (100) | - |
| Macrophage arc maximum, ˚ | 155.0 (118.5, 221.8) | 121.0 (84.0, 158.0) | 0.02 |
| Macrophage length, mm | 14.1 (10.2, 19.6) | 8.0 (5.0, 13.5) | 0.02 |
| Thrombus | 6 (50.0) | 5 (8.8) | < 0.01 |
| Plaque rupture | 4 (30.8) | 5 (8.8) | 0.06 |
| Calcium deposition | 11 (84.6) | 52 (91.2) | 0.39 |
| Calcium arc maximum, ˚ | 84.5 (76.8, 205.0) | 101.0 (67.0, 161.0) | 0.92 |
| **IVUS** | | | |
| EEM volume, mm$^3$ | 348.8 (193.4, 448.8) | 289.0 (189.7, 359.0) | 0.29 |
| TAV, mm$^3$ | 214.9 (126.0, 306.4) | 164.2 (94.9, 234.0) | 0.24 |
| PAV, % | 64.70 (60.35, 66.55) | 61.10 (54.25, 66.20) | 0.15 |
| Plaque burden, % | 77.86 ± 6.54 | 75.77 ± 7.25 | 0.34 |
| Plaque plus media, mm$^2$ | 7.80 (5.90, 9.40) | 7.60 (5.35, 9.75) | 0.95 |
| **NIRS** | | | |
| maxLCBI$_{lesion}$ | 158.2 (121.3, 256.6) | 85.1 (60.1, 121.8) | <0.01 |
| maxLCBI$_{4mm}$ | 519.7 (398.1, 721.0) | 370.3 (251.9, 472.9) | <0.01 |
| maxLCBI$_{10mm}$ | 379.9 (261.8, 560.1) | 244.0 (123.3, 339.6) | 0.01 |

Values are mean ± SD, median (interquartile range) or n (%); TCFA = thin-cap fibroatheroma; OCT = optical coherence tomography; CSA = cross-sectional area; IVUS = intravascular ultrasound; EEM = external elastic membrane; TAV = total atheroma volume; PAV = percent atheroma volume; NIRS = near-infrared spectroscopy; maxLCBI$_{lesion}$, maxLCBI$_{4mm}$, maxLCBI$_{10mm}$ = maximum lipid core burden index for the whole lesion and within 4-mm and 10-mm segments, respectively.

calcium and calcium arc were comparable between the groups. There were no differences in IVUS-defined TAV, PAV and plaque burden between lesions with and without TCFA. NIRS-defined maxLCBI$_{lesion}$, maxLCBI$_{10mm}$ and maxLCBI$_{4mm}$ were significantly higher in TCFA lesions.

## Wall shear stress calculation and thin-cap fibroatheroma

Angiography and CFD findings are summarized in Table 3. Mean WSS$_{proximal}$ was significantly higher in lesions with TCFA (10.20 [IQR: 5.01 to 16.93Pa]) compared to lesions without TCFA (5.84 [IQR: 3.74 to 8.29Pa], p = 0.02). Similarly, mean WSS in the total lesion (WSS$_{lesion}$) was higher in TCFA group (12.37 [IQR: 6.36 to 14.55Pa]) versus non-TCFA group (6.95 [IQR: 4.41 to 11.60Pa], p = 0.04). Percent diameter stenosis and lesion length were comparable between the groups. 3D contrast velocity in TCFA group (244±36mm/s) was higher than that in non-TCFA group (211±50mm/s, p = 0.03).

In univariate logistic regression analysis, mean WSS$_{proximal}$, lesion WSS and 3D contrast velocity were associated with the presence of TCFA (Table 4). After adjustments for the velocity, mean WSS$_{proximal}$ remained an independent predictor of OCT-defined TCFA (odds ratio:

**Table 3. Angiographically derived measurements and computational fluid dynamics findings.**

|  | TCFA (n = 13) | No TCFA (n = 57) | P-value |
|---|---|---|---|
| %DS | 52.2 ± 9.6 | 50.0 ± 8.2 | 0.40 |
| Lesion length, mm | 24.1 (13.0, 37.7) | 19.9 (14.1, 25.2) | 0.22 |
| 3D contrast velocity, m/s | 244 ± 36 | 211 ± 50 | 0.03 |
| Total lesion mean WSS, Pa | 12.37 (6.36, 14.55) | 6.95 (4.41, 11.60) | 0.04 |
| Upstream mean WSS, Pa | 3.42 (2.98, 5.49) | 2.69 (2.15, 4.44) | 0.25 |
| Proximal mean WSS, Pa | 10.20 (5.01, 16.93) | 5.84 (3.74, 8.29) | 0.02 |
| Middle mean WSS, Pa | 11.60 (4.82, 19.37) | 10.38 (5.20, 16.29) | 0.82 |
| Distal mean WSS, Pa | 5.67 (3.45, 6.40) | 4.15 (2.54, 7.69) | 0.61 |
| Downstream mean WSS, Pa | 3.85 (2.96, 8.41) | 4.30 (3.14, 7.11) | 1.00 |

Values are mean ± SD or median (interquartile range); TCFA = thin-cap fibroatheroma; DS = diameter stenosis; WSS = wall shear stress.

1.105; 95%CI: 1.007–1.213, p = 0.04). The optimal cutoff value of mean $WSS_{proximal}$ to predict TCFA was 6.79Pa (AUC: 0.71; sensitivity: 0.77; specificity: 0.63, p = 0.02) (S2 Fig). Fig 2A demonstrates a representative case with the computed mean $WSS_{proximal}$ of 16.50Pa. OCT-verified TCFA was detected in the middle segment. Fig 2B shows CFD analysis of a case without TCFA. Mean WSS in the proximal segment was 3.81Pa and fibroatheroma with FCT of 170μm was detected in the middle of the lesion. In contrast, the maximal WSS was not an independent predictor of TCFA lesion (S1 and S2 Tables).

## Discussion

The study evaluated the relationship between angiography-based WSS and the presence of OCT-verified TCFA in angiographically obstructive lesions in patients with stable CAD. The main findings of the study are: 1) Mean WSS in the proximal segment of the lesion was an independent predictor of TCFA; 2) the optimal cutoff value of $WSS_{proximal}$ to predict TCFA was 6.79Pa (AUC: 0.71; sensitivity: 0.77; specificity: 0.63, p = 0.02). In addition, TCFA lesions with high WSS had higher lipid and macrophage content assessed by OCT and NIRS. To our knowledge, the study is the first report showing association between angiography-based WSS and the presence of OCT-confirmed TCFA in obstructive coronary lesions.

During the development of atherosclerosis, the plaques expand outward to preserve luminal patency. When the plaques grow and the further outward expansion is limited, plaque

**Table 4. Relationship between TCFA and hemodynamics.**

|  | Odds ratio | 95%CI | P value |
|---|---|---|---|
| Total lesion mean WSS, Pa | 1.135 | 1.007–1.280 | 0.04 |
| Upstream mean WSS, Pa | 1.013 | 0.800–1.282 | 0.92 |
| Proximal mean WSS, Pa | 1.125 | 1.020–1.240 | 0.02 |
| Middle mean WSS, Pa | 1.021 | 0.944–1.123 | 0.61 |
| Distal mean WSS, Pa | 0.997 | 0.906–1.096 | 0.95 |
| Downstream mean WSS, Pa | 0.998 | 0.790–1.261 | 0.99 |
| 3D contrast velocity, mm/s | 1.014 | 1.001–1.028 | 0.04 |
| Total lesion mean WSS, Pa (adjusted for 3D contrast velocity) | 1.090 | 0.950–1.251 | 0.22 |
| Proximal mean WSS, Pa (adjusted for 3D contrast velocity) | 1.105 | 1.007–1.213 | 0.04 |

TCFA = thin-cap fibroatheroma; CI = confidence Interval, WSS = wall shear stress.

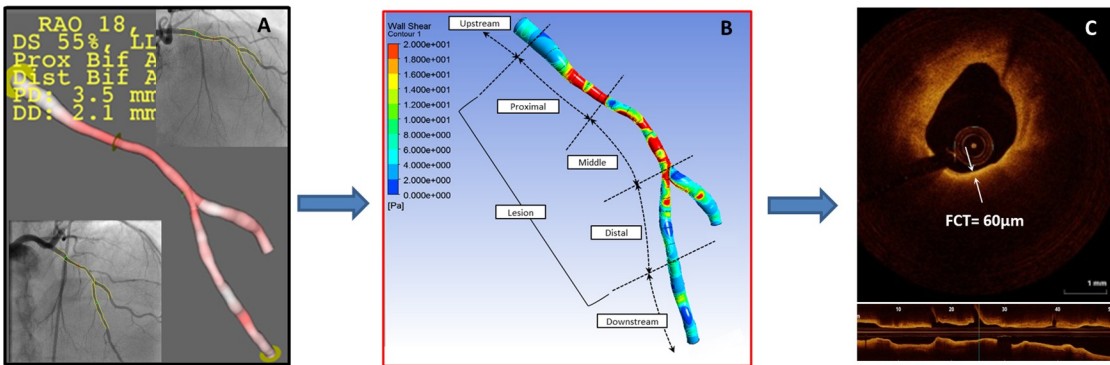

**Fig 2. High mean wall shear stress in the proximal segment of coronary lesions is associated with the presence thin-cap fibroatheroma verified by optical coherence tomography.** Color-coded WSS maps for two representative cases from patients with (A) thin-cap fibroatheroma lesion identified by OCT with minimal fibrous cap thickness 60 μm (A, inset) and WSS measured in the proximal segment of the lesion 16.5 Pa and (B) lesion with thick fibrous cap of 170 μm (B, inset) and proximal WSS 3.81 Pa. FCT = fibrous cap thickness; TCFA = thin-cap fibroatheroma; WSS = wall shear stress.

progression necessarily encroaches on the lumen and causes luminal narrowing [16, 17]. In the advanced stages of atherosclerosis, both low [18–20] and high [5–9, 21–26] wall shear stress have been associated with plaque vulnerability and clinical events. Although some MI may be caused by rupture of plaques with a mild degree of lumen narrowing, in the majority of cases there is a rapid progression of high-risk plaques with lumen narrowing before MI [17]. As plaques develop with lumen narrowing over time, high WSS occurs at the stenotic lesion. Experimental studies have shown that high WSS modifies gene expression and activates matrix metalloproteinases, which lead to increased inflammation, collagen and elastin degradation, vascular smooth muscle cell apoptosis and progression of the atheromatous lipid core [3, 27, 28]. In line with the experimental studies, IVUS study has demonstrated that high WSS was associated with development of necrotic core and regression of fibrous and fibro fatty tissue, suggesting transformation to more vulnerable plaques [5]. In addition, localized elevation of shear stress has been shown to be a trigger of fibrous cap rupture [6]. Consistent with our findings, high WSS was strongly associated with TCFA and lipid-rich plaques in an OCT based CFD analysis of intermediate stenoses [24]. Lipid-rich plaques detected by NIRS have been shown to co-localize with high WSS in an IVUS based CFD study in non-culprit lesions of patients presenting with ACs [23]. In a recent 3D QCA based CFD analysis, a combination of high WSS and high-risk plaque morphology provided an accurate identification of non-obstructive lesions responsible for the future events. Based on the recent findings, high WSS has been recently proposed as a possible causative factor promoting the development of high risk plaques responsible for the majority of acute coronary events. The association between high WSS and the presence of OCT-defined TCFA lesions with greater lipid arc and higher macrophage content observed in the study further supports the hypothesis for WSS as an independent mechanism underlying development of vulnerable plaques.

A post hoc analysis of FAME II trial had recently demonstrated that angiography-based high WSS in the proximal segments of obstructive lesions was a predictor of MI in patients with stable CAD. The intravascular imaging was not performed in the study [9]. TCFA is pathologically characterized by a necrotic core covered by a thin fibrous cap with numerous macrophages [29] and high-resolution OCT imaging can detect TCFA in vivo [30]. The present study demonstrated that the high WSS in the proximal segment was predictive of the presence of OCT-verified TCFA with additional vulnerable plaque features including a greater lipid arc, a longer lipid length and a higher LCBI by NIRS compared to the lesions without TCFA.

These multimodality imaging features of plaque vulnerability have been previously identified as predictors of adverse cardiovascular events [31, 32]. In addition, the TCFA lesions had higher content of macrophages, which contribute to plaque destabilization and rupture [33]. Taken together the findings of this and FAME II trial sub analysis provide an explanation for the association between high proximal WSS and myocardial infarction (Fig 2) [9].

In conclusion, angiography-based high WSS in the proximal segments of angiographically obstructive lesions was an independent predictor of OCT-verified TCFA. Intracoronary imaging including OCT and NIRS/IVUS are presently necessary to evaluate the plaque vulnerability, however, an additional invasive procedure is required to obtain the intracoronary images. The study demonstrates a possibility to detect high-risk plaques using angiography only combined with WSS assessment which can potentially be used in clinical practice.

There were several limitations in the study. First, the number of study population was small. Second, OCT and IVUS was not used to perform CFD and combined models might be more accurate, however what is novel in the study is that local WSS computed based on angiography which is usually performed in daily clinical practice was predictive for high-risk plaques in obstructive lesions in patients with CAD.

## Supporting information

**S1 Fig. Representative multimodality images of a study lesion.** OCT detected TCFA with minimal fibrous cap thickness 50 μm (inset) and a 305˚ lipid arc (A); large attenuated plaque detected by IVUS with maximal lipid core burden index (LCBI) within a 4 mm segment 465 detected by NIRS.
(DOCX)

**S2 Fig. Receiver-operating characteristic curve to predict thin-cap fibroatheroma lesions using angiography-based CFD analysis of wall shear stress in obstructive coronary lesions.**
(DOCX)

**S1 Table. Angiographically derived measurements and computational fluid dynamics: Maximal WSS.**
(DOCX)

**S2 Table. Relationship between TCFA and maximal WSS.**
(DOCX)

## Author Contributions

**Conceptualization:** Valentin Fuster, Habib Samady, Joseph Sweeny, Roxana Mehran, Jagat Narula, Annapoorna S. Kini.

**Data curation:** Yuliya Vengrenyuk.

**Formal analysis:** Naotaka Okamoto, Yuliya Vengrenyuk, Keisuke Yasumura.

**Investigation:** Valentin Fuster, Habib Samady, Usman Baber, Nitin Barman, Joseph Sweeny, Prakash Krishnan, Samin K. Sharma.

**Methodology:** Naotaka Okamoto, Usman Baber, Nitin Barman, Javed Suleman, Joseph Sweeny.

**Project administration:** Yuliya Vengrenyuk.

**Resources:** Javed Suleman, Prakash Krishnan.

**Software:** Naotaka Okamoto, Keisuke Yasumura.

**Supervision:** Valentin Fuster, Habib Samady, Nitin Barman, Prakash Krishnan, Samin K. Sharma, Annapoorna S. Kini.

**Validation:** Keisuke Yasumura, Javed Suleman.

**Visualization:** Jagat Narula.

**Writing – original draft:** Naotaka Okamoto, Yuliya Vengrenyuk.

**Writing – review & editing:** Roxana Mehran, Jagat Narula, Annapoorna S. Kini.

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
