## [Decision Letter · Decision Letter 0]

1 Oct 2020

PONE-D-20-22797

Relationship between high shear stress and OCT-verified thin-cap fibroatheroma in patients with coronary artery disease

PLOS ONE

Dear Dr Kini

Thank you for submitting your manuscript to PLOS ONE. After careful consideration, we feel that it has merit but does not fully meet PLOS ONE’s publication criteria as it currently stands. Therefore, we invite you to submit a revised version of the manuscript that addresses the points raised during the review process.

We look forward to receiving your revised manuscript.

Kind regards,

Xianwu Cheng, M.D., Ph.D., FAHA

Academic Editor

PLOS ONE

Additional Editor Comments:

Although the topic is interesting, as you will gather from the reviews, the referees identified substantive methodological problems, statistical analysis, and data presentation as well as the recruitment of all subjects.　 The editorial broad member also concurs. You may resubmit a revised version but it will be re-reviewed and there exists no guarantee that even with revision it will necessarily be accepted.

Journal Requirements:

"I have read the journal's policy and the authors of this manuscript have the following competing interests: Dr. Samady has received research grants from Abbott Vascular, Medtronic, National Institutes of Health, St. Jude Medical, and Gilead. Dr. Baber: speaker honoraria from Boston Scientific and Amgen, speaker honoraria and grants from AstraZeneca; Dr. Mehran: Abbott Vascular consultant and research grant, Boston Scientific consultant. Dr. Sharma: speaker honoraria from Abbott, Boston Scientific, Cardiovascular Systems, Inc. All other authors have no relationships relevant to the contents of this paper. "

Reviewers' comments:

Reviewer's Responses to Questions

**Comments to the Author**

1. Is the manuscript technically sound, and do the data support the conclusions?

Reviewer #1: Partly

Reviewer #2: Yes

2. Has the statistical analysis been performed appropriately and rigorously? 

Reviewer #1: I Don't Know

Reviewer #2: Yes

3. Have the authors made all data underlying the findings in their manuscript fully available?

Reviewer #1: Yes

Reviewer #2: Yes

4. Is the manuscript presented in an intelligible fashion and written in standard English?

Reviewer #1: Yes

Reviewer #2: Yes

5. Review Comments to the Author

Reviewer #1: This manuscript addresses an important question, the relationship between blood shear stress and thin cap fibroatheroma. The results are suggestive but the results are presented in a cursory manner with important details. Additional information is needed about the following:

1. The authors need to verify that they have performed the appropriate analysis to optimize the meshes for the numerical simulation and provide details on the mesh sizes and number of elements used in these validations. Indicate spatial resolution of shear stress.

2. Show representative inlet flow waveforms and explain how they were obtained.

3. Provide details on wall shear stress calculation over the plaque and how averaging over the cardiac cycle was performed. The lengths of lesion used to calculate proximal, middle and distal WSS need to be defined.

4. Since different methods were used to measure the dimensions of the fiberoatheromatous plaque and reconstruct the vessel geometry, describe the agreement in plaque dimensions using angiography and OCT and NIRS/IVUS.

5. Clarify whether the odds ratio means the likelihood of finding a fiberoatheromatous lesion based on a given shear stress or if a lesion is located in a region with a given proximal wall shear stress the odds of it being fiberoatheromatous.

A minor point to correct is the following.

Figure 2. Panels A, B, and C are not described.

Reviewer #2: The study aims to test the hypothesis that invasive coronary angiography-based high WSS is associated with the presence of TCFA detected by OCT in obstructive lesions. This manuscript selected patients who had two angiographic projections to create a 3D reconstruction model to assess WSS. The results demonstrated that high WSS in the proximal segments of obstructive lesions is an independent predictor of OCT-verified TCFA.

The paper is well written. Research was carefully designed and performed. This is a very important study and should be published with some minor changes.

1. Introduction.

The current manuscript is short of past publications on high WSS research. Some published studies on high WSS should be included and identify the research gaps that current research is trying to address.

2. Method.

For the convenience of understanding and reading, the definition of some plaque morphology parameters (such as LCBI etc.) should be expressed by some figures or formulas.

3. Result.

The study aims to test the hypothesis that invasive coronary angiography-based high WSS is associated with the presence of TCFA detected by OCT in obstructive lesions. Use of mean WSS is fine, but maximum WSS could also be considered which could be a more sensitive measure? There is a chance that mean WSS could get some important findings “averaged out”. The author needs to consider the maximum WSS to explain whether the mean WSS in the manuscript is a better measure to test the hypothesis?

4. Table3. Please give the explanation of DS (diameter stenosis).

5. “The optimal cutoff value of WSSproximal to predict TCFA was 6.79Pa (AUC: 0.710; sensitivity: 76.9%; specificity: 63.2%, p=0.02).” A figure with the ROC curve is needed to show where the values come from.

6. PLOS authors have the option to publish the peer review history of their article (what does this mean?). If published, this will include your full peer review and any attached files.

Reviewer #1: **Yes: **George A. Truskey

Reviewer #2: No

---

## [Author Response · Author response to Decision Letter 0]

5 Nov 2020

Response to Reviewers

Additional Editor Comments:

Although the topic is interesting, as you will gather from the reviews, the referees identified substantive methodological problems, statistical analysis, and data presentation as well as the recruitment of all subjects.　 The editorial broad member also concurs. You may resubmit a revised version but it will be re-reviewed and there exists no guarantee that even with revision it will necessarily be accepted.

 Journal Requirements:

Response: The manuscript was re-formatted according to PLOS ONE’s style requirements and file naming guidelines.

"I have read the journal's policy and the authors of this manuscript have the following competing interests: Dr. Samady has received research grants from Abbott Vascular, Medtronic, National Institutes of Health, St. Jude Medical, and Gilead. Dr. Baber: speaker honoraria from Boston Scientific and Amgen, speaker honoraria and grants from AstraZeneca; Dr. Mehran: Abbott Vascular consultant and research grant, Boston Scientific consultant. Dr. Sharma: speaker honoraria from Abbott, Boston Scientific, Cardiovascular Systems, Inc. All other authors have no relationships relevant to the contents of this paper. "

Response: The statement has been added to the cover letter for the revised manuscript 

Review Comments to the Author

Reviewer #1

This manuscript addresses an important question, the relationship between blood shear stress and thin cap fibroatheroma. The results are suggestive but the results are presented in a cursory manner with important details. Additional information is needed about the following:

1. The authors need to verify that they have performed the appropriate analysis to optimize the meshes for the numerical simulation and provide details on the mesh sizes and number of elements used in these validations. Indicate spatial resolution of shear stress.

Response: 

We greatly appreciate the reviewer’s comments and add additional information on mesh and solution convergence analyses, boundary conditions and WSS averaging to the revised manuscript as described below (pages 2-4).

The appropriate elements size for the models was determined as a results of a mesh refinement study to ensure that WSS values are size independent. The discrepancy in the mean WSS values for each segment of 3% of less was acceptable. The average length of the study vessels was 22 mm and the largest (proximal) diameter was 3.1 mm. As a result of the mesh convergence analyses, we used the following settings for mesh generating: max face size 0.2 mm, min size 0.02 mm. The average number of elements in the study was 287,265 ± 114,712 (surface area 0.021±0.010 mm2 and min edge length 0.044±0.026 mm). Each simulation was monitored for convergence using absolute criteria with 10-6 as a threshold for the residual error. Several models required local refinement of the mesh at the bifurcation segment in order to achieve convergent solution. We added the following details to the revised manuscript:

Methods, page 6: The optimal mesh size was determined based on a mesh refinement study to ensure that the results are not affected by the element size. The average number of finite elements in the study was 287,265, and the mean surface area was 0.021 mm2.

2. Show representative inlet flow waveforms and explain how they were obtained.

Response: We calculated velocity of the contrast in angiography images for each vessel using a frame count method with the first frame showing the front of contrast bolus entering analyzed segment and the last frame in which contrast reaches the end of the segment. The velocity was calculated as follows: 

[segment length assessed by angiography, mm]*[acquisition speed, frames/sec]/[last frame number – first frame number, frames] as described in the Fame II post-hoc analysis by Kumar and co-authors (Ref #9). Similar to the study, the patient-specific calculated velocity was applied as an inlet boundary condition with a flat profile. We added the following details to the revised manuscript:

Materials and Methods (Page 6, new text is underlined): Patient-specific inlet velocities were calculated using flame counts ([segment length assessed by angiography]*[acquisition speed]/[last frame number – first frame number, frames]) and applied as inlet boundary conditions with a flat profile as previously described (9).

3. Provide details on wall shear stress calculation over the plaque and how averaging over the cardiac cycle was performed. The lengths of lesion used to calculate proximal, middle and distal WSS need to be defined.

Response:

CFD simulations with the pulsatile flow were not performed in the study, blood flow was assumed to be steady. Each angiography-defined lesion was divided into 3 equal parts: proximal, middle and distal in the same manner as in the FAME II sub study (ref #9), since we wanted to test the hypothesis that the high WSS is associated with vulnerable plaques by OCT to provide an explanation for the association of high WSS in the proximal segment with future MIs reported in the study. In the original version of the manuscript, we calculated mean WSS in the whole lesion and each sub segment. For the revised paper, we performed additional analyses for the maximal WSS per suggestion of the Reviewer #2, but the mean WSS remained the most sensitive predictor of the presence of TCFA lesion. Considering that the average lesion length in our study was 22 mm (Table 3), each lesion sub segment was around 7.3 mm depending on the case. We’ve added the following clarification to the revised manuscript (page 5, new text is underlined):

Materials and Methods (Page 6, new text is underlined): No-slip boundary condition was applied at the vessel wall and blood flow was assumed to be steady Each angiography-defined lesion was divided into 3 equal parts: proximal, middle and distal following the segment definitions introduced in the WSS FAME II sub-study (9).

4. Since different methods were used to measure the dimensions of the fiberoatheromatous plaque and reconstruct the vessel geometry, describe the agreement in plaque dimensions using angiography and OCT and NIRS/IVUS.

Response: The vessel geometries were reconstructed based on angiography images using two end-diastolic projections more than 25 degrees apart using validated software QAngio XA 3D. Intravascular imaging (OCT and NIRS/IVUS) was used only to characterize plaque morphology for each lesion. Anatomical landmarks (side branches) visible in both angiography and OCT images were used to identify the location of TCFA lesions within the angiography-based models. 

5. Clarify whether the odds ratio means the likelihood of finding a fiberoatheromatous lesion based on a given shear stress or if a lesion is located in a region with a given proximal wall shear stress the odds of it being fiberoatheromatous.

Response: We apologize for the confusion. Our data showed the association of high WSS with the presence of OCT-defined TCFA rather than the co-localization of the two measures. While we did mention this in the abstract, it was not clearly described throughout the text. We corrected the wording in several places including pages 3, 11, 13, 15 as follows : ”The study evaluated the relationship between angiography-based WSS and the presence of OCT-verified TCFA in angiographically obstructive lesions in patients with stable CAD”

A minor point to correct is the following.

Figure 2. Panels A, B, and C are not described.

Response: We added Figure 2 panels’ description to the revised manuscript and re-arranged Figures 1 and 2 to better describe the study methods and results. 

Reviewer #2

The study aims to test the hypothesis that invasive coronary angiography-based high WSS is associated with the presence of TCFA detected by OCT in obstructive lesions. This manuscript selected patients who had two angiographic projections to create a 3D reconstruction model to assess WSS. The results demonstrated that high WSS in the proximal segments of obstructive lesions is an independent predictor of OCT-verified TCFA.

The paper is well written. Research was carefully designed and performed. This is a very important study and should be published with some minor changes.

1. Introduction.

The current manuscript is short of past publications on high WSS research. Some published studies on high WSS should be included and identify the research gaps that current research is trying to address.

Response: We greatly appreciate the reviewer’s comments and add some of the most recent reports on the association of high WSS with lesion vulnerability (please see below). While majority of the studies used intravascular imaging or CTA to create 3D CFD models (1-5), 3D-QCA-derived high WSS in combination with lesion morphology was able to identify plaques prone to future MACE in the last month report from Bourantas and co-authors (6). 

1. Hartman EMJ, De Nisco G, Kok AM et al. Lipid-rich Plaques Detected by Near-infrared Spectroscopy Are More Frequently Exposed to High Shear Stress. Journal of cardiovascular translational research. 2020. Epub 2020/10/10.

2. Toba T, Otake H, Choi G et al. Wall Shear Stress and Plaque Vulnerability: Computational Fluid Dynamics Analysis Derived from cCTA and OCT. JACC Cardiovascular imaging. 2020. Epub 2020/09/21.

3. Lee JM, Choi G, Koo BK et al. Identification of High-Risk Plaques Destined to Cause Acute Coronary Syndrome Using Coronary Computed Tomographic Angiography and Computational Fluid Dynamics. JACC Cardiovascular imaging. 2019;12(6):1032-43. Epub 2018/03/20.

4. Yamamoto E, Thondapu V, Poon E et al. Endothelial Shear Stress and Plaque Erosion: A Computational Fluid Dynamics and Optical Coherence Tomography Study. JACC Cardiovascular imaging. 2019;12(2):374-5. Epub 2018/10/22.

5. Murata N, Hiro T, Takayama T et al. High shear stress on the coronary arterial wall is related to computed tomography-derived high-risk plaque: a three-dimensional computed tomography and color-coded tissue-characterizing intravascular ultrasonography study. Heart and vessels. 2019;34(9):1429-39. Epub 2019/04/13.

6. Bourantas CV, Zanchin T, Torii R et al. Shear Stress Estimated by Quantitative Coronary Angiography Predicts Plaques Prone to Progress and Cause Events. JACC Cardiovascular imaging. 2020;13(10):2206-19. Epub 2020/05/18.

In addition, the following summary was included in the revised Discussion (page 14): “Consistent with our findings, high WSS was strongly associated with TCFA and lipid-rich plaques in an OCT based CFD analysis of intermediate stenoses [24]. Lipid-rich plaques detected by NIRS have been shown to co-localize with high WSS in an IVUS based CFD study in non-culprit lesions of patients presenting with ACs [23]. In a recent 3D QCA based CFD analysis, a combination of high WSS and high-risk plaque morphology provided an accurate identification of non-obstructive lesions responsible for the future events.”

2. Method. For the convenience of understanding and reading, the definition of some plaque morphology parameters (such as LCBI etc.) should be expressed by some figures or formulas.

Response: We appreciate the suggestion and provide definitions for the imaging characteristics used in the study in the Methods and a new Supporting Figure 1 (S1 Fig) showing representative images of a lesion acquired with different image modalities.

Materials and Methods. Page 4, 5 (new text is underlined): “Lipid core was identified as a signal-poor region with poorly delineated borders, little or no signal backscattering, and an overlying signal-rich layer, the fibrous cap (S1 Fig) … Raw spectroscopic information from NIRS chemograms was transformed into a probability of lipid that was mapped to a red-to-yellow colour scale, with a low probability of lipid shown as red and a high probability of lipid shown as yellow (S1 Fig). Lipid core burden index (LCBI) was calculated by dividing the number of yellow pixels (probability ≥0.6) by the total number of viable pixels within the plaque multiplied by 1000.”

S1 Figure. Representative multimodality images of a study lesion: OCT detected TCFA with minimal fibrous cap thickness 50 µm (inset) and a 305° lipid arc (A); large attenuated plaque detected by IVUS with maximal lipid core burden index (LCBI) within a 4 mm segment 465 detected by NIRS (yellow area). 

3. Result.

The study aims to test the hypothesis that invasive coronary angiography-based high WSS is associated with the presence of TCFA detected by OCT in obstructive lesions. Use of mean WSS is fine, but maximum WSS could also be considered which could be a more sensitive measure? There is a chance that mean WSS could get some important findings “averaged out”. The author needs to consider the maximum WSS to explain whether the mean WSS in the manuscript is a better measure to test the hypothesis?

Response: Thank you for the interesting suggestion. We performed additional analyses to test whether the maximal WSS would be a more sensitive predictor of the presence of TCFA compared to the mean WSS. The results are shown in new S1 and S2 Tables and below. While proximal WSS (max) was significantly higher in TCFA lesions compared to non TCFA lesions (S1 Table), it was not an independent predictor of TCFA by univariate logistic regression analysis (S2 Table) with or without adjustment for 3D contrast velocity.

 Supporting Table 1. Angiographically derived measurements and computational fluid dynamics: maximal WSS 

 TCFA (n=13) No TCFA (n=57) P-value

Total lesion WSS (max), Pa 65.8 (31.3, 98.7) 44.2 (23.7, 67.6) 0.04

Upstream WSS (max), Pa 8.5 (6.2, 21.5) 8.7 (4.4, 21.7) 0.63

Proximal WSS (max), Pa 34.6 (14.7, 54.5) 13.6 (8.6, 29.6) 0.04

Middle WSS (max), Pa 60.5 (25.1, 91.3) 34.9 (18.9, 58.8) 0.12

Distal WSS (max), Pa 24.6 (13.3, 48.9) 16.4 (8.8, 34.0) 0.25

Downstream WSS (max), Pa 14.7 (6.0, 42.7) 13.8 (7.0, 30.1) 0.88

Values are mean ± SD or median (interquartile range); TCFA= thin-cap fibroatheroma; DS= diameter stenosis; WSS= wall shear stress.

Supporting Table 2. Relationship between TCFA and maximal WSS 

 Odds ratio 95%CI P value

Total lesion WSS (max), Pa 1.017 1.001-1.034 0.04

Upstream WSS (max), Pa 0.996 0.958-1.035 0.84

Proximal WSS (max), Pa 1.014 0.993-1.036 0.20

Middle WSS (max), Pa 1.013 0.995-1.032 0.16

Distal WSS (max), Pa 1.010 0.990-1.031 0.32

Downstream WSS (max), Pa 0.903 0.968-1.038 0.90

Total lesion WSS (max), Pa (adjusted for 3D contrast velocity) 1.012 0.988-1.035 0.33

Proximal WSS (max), Pa (adjusted for 3D contrast velocity) 1.015 0.992-1.037 0.83

4. Table3. Please give the explanation of DS (diameter stenosis).

Response: Percent diameter stenosis or diameter stenosis is defined as the difference between the reference vessel diameter and minimal luminal diameter, divided by the reference and multiplied by 100. All the measurements were performed automatically by QAngio XA 3D RE software in the study. We added the following clarification to the revised manuscript: 

Page 5 (new text is underlined): “Three-dimensional quantitative coronary angiography (3D-QCA) and vessel reconstructions were performed using two end-diastolic projections at least 25° apart with commercially available software QAngio XA 3D RE (Medis, Leiden, The Netherlands) followed by automatic quantification of 3D lesion length, minimal and reference diameters, and percent diameter stenosis (DS) (Figure 1A) (14)”. 

5. “The optimal cutoff value of WSSproximal to predict TCFA was 6.79Pa (AUC: 0.710; sensitivity: 76.9%; specificity: 63.2%, p=0.02).” A figure with the ROC curve is needed to show where the values come from.

Response: We apologize for the oversight, the ROC figure below was added to the revised manuscript 

6. PLOS authors have the option to publish the peer review history of their article (what does this mean?). If published, this will include your full peer review and any attached files.

Do you want your identity to be public for this peer review? For information about this choice, including consent withdrawal, please see our Privacy Policy.

Reviewer #1: Yes: George A. Truskey

Reviewer #2: No

---

## [Decision Letter · Decision Letter 1]

23 Nov 2020

PONE-D-20-22797R1

Relationship between high shear stress and OCT-verified thin-cap fibroatheroma in patients with coronary artery disease

PLOS ONE

Dear Dr Kini

Thank you for submitting your manuscript to PLOS ONE. After careful consideration, we feel that it has merit but does not fully meet PLOS ONE’s publication criteria as it currently stands. Therefore, we invite you to submit a revised version of the manuscript that addresses the points raised during the review process.

We look forward to receiving your revised manuscript.

Kind regards,

Xianwu Cheng, M.D., Ph.D., FAHA

Academic Editor

PLOS ONE

Reviewers' comments:

Reviewer's Responses to Questions

**Comments to the Author**

1. If the authors have adequately addressed your comments raised in a previous round of review and you feel that this manuscript is now acceptable for publication, you may indicate that here to bypass the “Comments to the Author” section, enter your conflict of interest statement in the “Confidential to Editor” section, and submit your "Accept" recommendation.

Reviewer #1: (No Response)

Reviewer #2: All comments have been addressed

2. Is the manuscript technically sound, and do the data support the conclusions?

Reviewer #1: Yes

Reviewer #2: (No Response)

3. Has the statistical analysis been performed appropriately and rigorously? 

Reviewer #1: Yes

Reviewer #2: (No Response)

4. Have the authors made all data underlying the findings in their manuscript fully available?

Reviewer #1: Yes

Reviewer #2: (No Response)

5. Is the manuscript presented in an intelligible fashion and written in standard English?

Reviewer #1: Yes

Reviewer #2: (No Response)

6. Review Comments to the Author

Reviewer #1: The issues raised in the review of the original manuscript were addressed. There are a few minor details to address.

Specific Comments

p. 6, line 7, “flame counts”. Should be “frame counts”

p. 6, lines 4-15 “Mean and maximal WSS was calculated …” should be “Mean and

maximal WSS were calculated …”.

Table 2. Explain why the lipid arc maximum is presented rather than the average lipid arc.

Reviewer #2: (No Response)

7. PLOS authors have the option to publish the peer review history of their article (what does this mean?). If published, this will include your full peer review and any attached files.

Reviewer #1: **Yes: **George A Truskey

Reviewer #2: No

---

## [Author Response · Author response to Decision Letter 1]

23 Nov 2020

Reviewer #1: The issues raised in the review of the original manuscript were addressed. There are a few minor details to address.

Specific Comments

p. 6, line 7, “flame counts”. Should be “frame counts”

p. 6, lines 4-15 “Mean and maximal WSS was calculated …” should be “Mean and

maximal WSS were calculated …”.

Response: We corrected both typos in the revised manuscript, thank you. 

Table 2. Explain why the lipid arc maximum is presented rather than the average lipid arc.

Response: We apologize for the oversight; the average lipid arc has been added to the Table 2 of the revised manuscript 

Reviewer #2: (No Response)

---

## [Decision Letter · Decision Letter 2]

2 Dec 2020

Relationship between high shear stress and OCT-verified thin-cap fibroatheroma in patients with coronary artery disease

PONE-D-20-22797R2

Dear Dr. Kini

We’re pleased to inform you that your manuscript has been judged scientifically suitable for publication and will be formally accepted for publication once it meets all outstanding technical requirements.

Kind regards,

Xianwu Cheng, M.D., Ph.D., FAHA

Academic Editor

PLOS ONE

Additional Editor Comments (optional):

None.

Reviewers' comments:

Reviewer's Responses to Questions

**Comments to the Author**

1. If the authors have adequately addressed your comments raised in a previous round of review and you feel that this manuscript is now acceptable for publication, you may indicate that here to bypass the “Comments to the Author” section, enter your conflict of interest statement in the “Confidential to Editor” section, and submit your "Accept" recommendation.

Reviewer #1: All comments have been addressed

Reviewer #2: All comments have been addressed

2. Is the manuscript technically sound, and do the data support the conclusions?

Reviewer #1: (No Response)

Reviewer #2: (No Response)

3. Has the statistical analysis been performed appropriately and rigorously? 

Reviewer #1: (No Response)

Reviewer #2: (No Response)

4. Have the authors made all data underlying the findings in their manuscript fully available?

Reviewer #1: (No Response)

Reviewer #2: (No Response)

5. Is the manuscript presented in an intelligible fashion and written in standard English?

Reviewer #1: (No Response)

Reviewer #2: (No Response)

6. Review Comments to the Author

Reviewer #1: (No Response)

Reviewer #2: (No Response)

7. PLOS authors have the option to publish the peer review history of their article (what does this mean?). If published, this will include your full peer review and any attached files.

Reviewer #1: **Yes: **George A. Truskey

Reviewer #2: No

---

## [Editor Report · Acceptance letter]

9 Dec 2020

PONE-D-20-22797R2 

Relationship between high shear stress and OCT-verified thin-cap fibroatheroma in patients with coronary artery disease 

Dear Dr. Kini:

I'm pleased to inform you that your manuscript has been deemed suitable for publication in PLOS ONE. Congratulations! Your manuscript is now with our production department. 

Kind regards, 

on behalf of

Associate Prof. Xianwu Cheng 

Academic Editor

PLOS ONE